

# An algorithm for discovering Lagrangians automatically from data

Daniel J.A. Hills, Adrian M. Grütter and Jonathan J. Hudson

Department of Physics, Imperial College of Science, Technology and Medicine, London, United Kingdom

## ABSTRACT

An activity fundamental to science is building mathematical models. These models are used to both predict the results of future experiments and gain insight into the structure of the system under study. We present an algorithm that automates the model building process in a scientifically principled way. The algorithm can take observed trajectories from a wide variety of mechanical systems and, without any other prior knowledge or tuning of parameters, predict the future evolution of the system. It does this by applying the principle of least action and searching for the simplest Lagrangian that describes the system's behaviour. By generating this Lagrangian in a human interpretable form, it can also provide insight into the workings of the system.

## INTRODUCTION

Modern science is, in many senses, highly automated. Experiments are frequently run under computer control, with data often recorded by the computer directly. Computerised data analysis and visualisation are widely used to process the resulting large volumes of data. Indeed, the ability to collect and analyse massive data sets is opening up an entirely new measure-first-ask-questions-later approach to science: the Square Kilometer Array radio telescope is expected to collect approximately one exabyte of data per day (*Newman & Tseng, 0000*); over $10^{14}$ collisions from the ATLAS detector were analysed in the search for the Higgs boson (*ATLAS collaboration, 2012*); and state-of-the-art whole-genome sequencers can currently sequence 600 gigabases per day (*Hayden, 2014*). In each of these examples the scientific questions are not fully formulated in advance of taking the data, and the question of how to best extract knowledge from the dataset is of great interest. This motivates the study of how to scale up the processes of scientific reasoning to take advantage of the wealth of available data.

Thus far, scientific reasoning has largely resisted automation. Hypothesising and refining models is still on the whole carried out by humans, with little direct support from computers. It has long been a desire of artificial intelligence researchers to automate this part of science, and with the growing volume of data available from experiments the motivation for this desire comes ever more sharply into focus. In this paper we present a step in this direction: an algorithm that automates finding a mathematical model for a system, in a scientifically principled way, by examining only its observed behaviour.

Corresponding author
Jonathan J. Hudson,
jony.hudson@imperial.ac.uk

Early attempts to automatically model physical systems searched for simple mathematical regularities in observed quantities. *Langley*'s (*1979*) BACON system was able to re-discover many simple laws—the ideal gas law, Ohm's law, Coulomb's law and others—from experimental data. *Dzeroski & Todorovski (1993)* went beyond simple static laws with their LAGRANGE system which was able to search for differential equations that governed observed time series. They extended this work to the LAGRAMGE system which additionally allowed an expert user to provide domain knowledge, improving the quality of the results (*Todorovski & Dzeroski, 1997*). The PRET system, developed by Bradley and collaborators (*2001*), brings to bear a variety of advanced AI techniques on the problem of identifying system differential equations. It has a sophisticated method for representing qualitative observations, and allows expert-user domain knowledge to be combined with automatic search very effectively. *Schmidt & Lipson (2009)* used a genetic programming approach to automatically evolve invariant mathematical expressions from observed data.[1] In the context of engineering, there is a significant body of work on 'system identification', with techniques ranging from very general ad hoc fitting methods to fitting detailed physical models representing important classes of system (*Sjberg et al., 1995*; *Ljung, 2010*).

In this work we take a different approach than those described above, the essence of which is that we embed a simple, general physical principle—the principle of least action—and very little else into our algorithm. While we are embedding the domain knowledge of a physicist in our algorithm, we are not embedding information about any particular physical system or class thereof. Rather we are capturing a deep understanding that has been distilled by physicists over the past 270 years, and packaging it into an algorithm that can be applied by non-experts. We find the algorithm to be surprisingly powerful, given its simplicity, but this power comes not from the ingenuity of its construction, rather from the broad applicability of the physical principle embedded in it.

## THE PRINCIPLE OF LEAST ACTION

The principle of least action is one of the most fundamental and most celebrated principles in physics. First proposed by *Maupertuis (1744)* and *Euler (1744)* it states that the problem of predicting the behaviour of many physical systems can be cast as finding the behaviour that minimises the expenditure given by some cost function. The total expenditure of the system is known as the action. It is a remarkable fact that the behaviour of a very wide range of physical systems—including those studied in classical mechanics, special and general relativity, quantum field theory, and optics—can all have their behaviour explained in terms of minimising a cost function.

Each physical system has its own cost function, and once this function is known it is possible to predict exactly what the system will do in the future. The exercise of determining the cost function—often known as Lagrange's function, or just the Lagrangian—for a particular physical system is central to physics. Feynman described this process well (*Feynman, Leighton & Sands, 1963*) as "some kind of trial and error" advising students that "You just have to fiddle around with the equations that you know and see if you can get them into the form of the principle of least action." In this paper we

[1] Note that Schmidt and Lipson originally claim that their technique is capable of discovering Lagrangians, but it has been shown that this is false except for Lagrangians of a very particular, trivial form (*Hillar & Sommer, 2012*). *Schmidt & Lipson (2010)* do not include their claim in a subsequent paper on the same work.

present an algorithm that does this "fiddling" automatically, without requiring the user to have any expertise in physics.

The Lagrangian is well-suited to be the output of an automated modelling algorithm. It has the desirable property in that it is a single, scalar expression that contains everything necessary to predict the system's future evolution. Consider, in contrast, finding the Hamiltonian where it would also be necessary to find the corresponding conjugate momenta. The Lagrangian has the additional quality that it is coordinate-independent and as a result can be written in any coordinate system. This is useful in the case of an automated algorithm where it is not obvious in which coordinate system the data might be presented. However, it should be noted that not all physical systems can be described by a Lagrangian. In particular, dissipative systems can not be modelled this way. Nevertheless, many interesting processes do admit a Lagrangian formulation, and what's more, as the algorithm is automatic little is lost by speculatively applying it to a system's trajectory. We might imagine a future where an ensemble of algorithms such as this one try to find an appropriate model for a system, based on a variety of physical principles and insights. We present this algorithm as a step towards such a future system.

The problem of taking a Lagrangian and automatically calculating the resulting motion of the system has been widely studied and applied. To the best of our knowledge, this work is the first that solves the inverse problem of taking the observed motion and calculating the Lagrangian for non-trivial systems.

## THE ALGORITHM

To find a model for a system we search over a space of possible Lagrangians. To do this we need three elements: an objective to guide the search, which will take the form of a score function; a representation of the possible Lagrangians; and an algorithm to execute the search over the possible Lagrangians, working to improve the score. We will first describe the score function, which is the central idea of the algorithm.

### Score function

The objective of our algorithm is to find a Lagrangian that, when integrated along the system's observed trajectory, yields a smaller total (action) than when integrated along any neighbouring trajectory. It would be possible to implement this definition of the least action principle directly in an algorithm, but instead we take an indirect approach that is more computationally efficient. For a Lagrangian $\mathcal{L}(\theta, \phi, \ldots, \dot{\theta}, \dot{\phi}, \ldots)$ and a trajectory $(\theta(t), \phi(t), \ldots, \dot{\theta}(t), \dot{\phi}(t), \ldots)$ it is possible to write down a condition, in the form of a set of differential equations, that must be satisfied if the action is to be stationary along the trajectory. These differential equations are known as Euler–Lagrange's equations,

$$\frac{d}{dt}\frac{\partial \mathcal{L}}{\partial \dot{q}} - \frac{\partial \mathcal{L}}{\partial q} = 0 \,,$$

where $q \in \{\theta, \phi, \ldots\}$. It is to be understood that the partial derivatives are taken symbolically with respect to the coordinates and velocities, which are then replaced with the time-dependent functions from the trajectory before the time-derivative is taken. We

can define a score function based on these conditions,

$$\text{EL}(\mathcal{L}) = \sum_{q \in \{\theta, \phi, \dots\}} \int \left( \frac{d}{dt} \frac{\partial \mathcal{L}}{\partial \dot{q}} - \frac{\partial \mathcal{L}}{\partial q} \right)^2 dt, \tag{1}$$

which is zero if the Euler–Lagrange equations are exactly satisfied. We note that Hillar and Sommer first proposed using a (different) score function derived from the Euler–Lagrange equations in *Hillar & Sommer (2012)*, but did not apply it to finding Lagrangians from data.

In practice our observations of the system are not functions $(\theta(t), \phi(t), \dots, \dot{\theta}(t), \dot{\phi}(t), \dots)$ but discretely sampled time-series of the coordinates and generalized velocities. The algorithm operates with a dataset which is a time-series of samples

$$D = ((\theta(1), \phi(1), \dots, \dot{\theta}(1), \dot{\phi}(1), \dots), \dots).$$

where time runs from $t = 1 \dots N$. The velocity samples may be either directly measured or derived from measured coordinate data. We divide this time-series into two portions, a training set, comprising samples $1 \dots M$, and a validation set of samples $M + 1 \dots N$. The algorithm will conduct its search using only the training set, reserving the validation set for out-of-sample measurement of the prediction error. In this way we can truly test the algorithm's ability to predict the future dynamics of the system. In all of the examples in this paper the sampling times will be evenly spaced, but this is not a requirement. We can discretize the Euler–Lagrange score function (Eq. (1)) to work with these sampled datasets, giving

$$\text{EL}_D(\mathcal{L}) = \sum_{t=1}^{M} \sum_{q \in \{\theta, \phi, \dots\}} \left( \left[ \frac{d}{dt} \frac{\partial \mathcal{L}}{\partial \dot{q}} \right]_t - \left[ \frac{\partial \mathcal{L}}{\partial q} \right]_t \right)^2, \tag{2}$$

where the subscript on the score indicates that it is taken with respect to the dataset $D$. The square-bracketed quantities in this expression are time-series, and the subscript indicates taking the element in this time-series at the given time. So, for instance, the first term in (2) is to be calculated, in principle, by: first differentiating the candidate Lagrangian $\mathcal{L}$ symbolically with respect to the appropriate generalized velocity; evaluating this quantity at every time-step in the dataset to yield a new time-series; taking the discrete derivative of this new time-series with respect to time; and finally finding the element at time $t$ in this time-derivative time-series. In practice, as we shall see below, a more computationally efficient implementation may be used.

The function $\text{EL}_D$ is the basis of the score function, capturing the principle of least action, but it is not sufficient on its own. While it is true that the Lagrangian we seek minimises $\text{EL}_D$, the converse is not true as there are other functions which minimise $\text{EL}_D$ but are not physically meaningful Lagrangians. The first class of functions that we wish to avoid are those which are numerically tiny, for instance $\mathcal{L} = 10^{-100}\theta$. We deal with these by

introducing a normalisation score for each candidate Lagrangian,

$$N_D(\mathcal{L}) = \sum_{t=1}^{M} \sum_{q \in \{\theta, \phi, \ldots\}} \left[ \frac{d}{dt} \frac{\partial \mathcal{L}}{\partial \dot{q}} \right]_t^2 + \left[ \frac{\partial \mathcal{L}}{\partial q} \right]_t^2.$$

We will compose our final score from the scores $\mathrm{EL}_D$ and $N_D$ in such a way, to be detailed below, that to score well a candidate Lagrangian must simultaneously have a low score for $\mathrm{EL}_D$ and a score of around one for $N_D$. The target value of one for $N_D$ is chosen arbitrarily. We can always arrange for the normalisation score to be approximately one, as the least-action trajectory is unchanged if the Lagrangian is scaled by a constant.

There is a second, more interesting class, of unwanted expressions that minimise the Euler–Lagrange score $\mathrm{EL}_D$. Consider, for instance, the candidate Lagrangian $\mathcal{L} = \theta^n \dot{\theta}$. This Lagrangian satisfies the Euler–Lagrange equations trivially, in a way that does not depend on the trajectory. Such path-independent least-action Lagrangians are interesting from a physics point-of-view, being closely related to gauge invariance, but here they are a nuisance. To guide the search away from these expressions we introduce a second 'control' trajectory, $C$. This trajectory is unrelated to the behaviour of the system under study and serves solely to eliminate path-independent Lagrangians. We reason that the Lagrangian that we are seeking will score well with $\mathrm{EL}_D$ but should score poorly on $\mathrm{EL}_C$, which is the Euler–Lagrange score evaluated along the control trajectory. The exact form of the control trajectory is unimportant so long as it not a valid trajectory of the system under study. In this work we use a control trajectory which is uniform motion in each coordinate, with velocity arbitrarily chosen to be 0.1, for all experiments.

We combine the three parts described above to give the search score function,

$$S(\mathcal{L}) = U(\mathrm{N}_D(\mathcal{L}))\, U(\mathrm{EL}_C(\mathcal{L})) \frac{\mathrm{EL}_D(\mathcal{L}) + \epsilon}{\mathrm{EL}_C(\mathcal{L}) + \epsilon}, \tag{3}$$

where $U(x) = \ln(x + \epsilon)^2 + 1$ is a function that is minimised, with value approximately one, when the argument is one. The small constant $\epsilon$, typically set to be $10^{-10}$, ensures that the score function has the desired asymptotic behaviour for small values of the numerator and denominator, even when faced with errors from finite precision machine numbers. The factor $U(\mathrm{EL}_C(\mathcal{L}))$ prevents the search algorithm from driving towards Lagrangians that perform badly on the real dataset, but even worse on the control data. Overall, the score function drives the search to find Lagrangians that simultaneously minimise the action along the observed trajectory while having a non-zero action along the control trajectory, and a normalisation score close to one. We note that the way that the score function is assembled is somewhat *ad hoc*. Its purpose is to guide the search to the correct answer while avoiding pathologies, and there are a number of ways this could be done—indeed, many were tried during development. This score function is simply presented as a particular arrangement that we have demonstrated to work.

Note that the score function, $S$, does not in any way consider whether the prediction of the candidate Lagrangian agrees with the training data. It only considers whether the trajectory satisfies a least action principle for the candidate Lagrangian. The fact that this,

on the face of it unrelated, objective leads to successful predictions is the insight from physics that we have embedded in our algorithm.

## Representation and search

We have experimented with two representations of candidate Lagrangians. The first, a restricted polynomial representation, allows a fast search algorithm to be implemented. It is limited in the Lagrangians it can represent exactly, although through Taylor's theorem it can find approximations to any Lagrangian. This representation was used to generate the bulk of the results in this paper, and we describe it in detail in this section. The second representation lifts some of the constraints of the restricted polynomial model, at the expense of vastly increased computational cost. We describe it in 'Generalisation'.

The restricted polynomial representation assumes that the Lagrangian can be represented by a polynomial in the coordinates and velocities. The model is a sum of monomial terms, parameterised by coefficients multiplying every term. We restrict this polynomial in two ways: we limit the maximum power of any coordinate or velocity to be $m$; and we limit the maximum degree of any combination of coordinates and velocities to $p$. In addition, we remove any terms from the model that can have no physical significance, that is terms that are constant or of the form $q^n \dot{q}$. These terms simply "fall through" the Euler–Lagrange equation without changing the resulting equations of motion, so there is no value in including them in the search. For example, for one variable $\theta$ with $m = 3$ and $p = 4$ the resulting model would be

$$c_1 \dot{\theta}^2 + c_2 \dot{\theta}^3 + c_3 \theta + c_4 \theta \dot{\theta}^2 + c_5 \theta \dot{\theta}^3 + c_6 \theta^2 + c_7 \theta^2 \dot{\theta}^2 + c_8 \theta^3.$$

For a given restricted polynomial model the score function is minimised by adjusting the parameters $c_i$. We conduct this optimisation using the Nelder–Mead simplex algorithm (*Nelder & Mead, 1965*) using the modified parameters of *Gao & Han (2012)* which improve the efficiency in high dimensions. The coefficients are bounded between $-1$ and $1$, enforced by a penalty function. We use a tight convergence tolerance, usually one part in $10^{10}$, to encourage the search to break out of local minima. We impose a maximum iteration limit, usually $5 \times 10^6$, on the search to ensure that it is bounded in time.

We do not know in advance what values of $m$ and $p$ are needed to accurately represent the Lagrangian of the system under study. What's more, we wish to find the simplest Lagrangian such that the trajectory satisfies the principle of least action. We approach this using a simple heuristic algorithm. We start with the smallest non-trivial model ($m = 2, p = 2$) and optimise the parameters with the simplex search. We then make an in-sample prediction of how well the optimised Lagrangian predicts the dynamics of the system in the training sample. This is done by generating equations of motion from the optimised Lagrangian and numerically solving them, using initial conditions derived from the first sample in the dataset. If this in-sample prediction fits better than a specified tolerance then we stop and return the Lagrangian. If it does not fit then we generate a larger model (i.e., with larger values of $m$ and/or $p$) and try again. The models are stepped through in increasing number of monomial terms. This proceeds until either a model

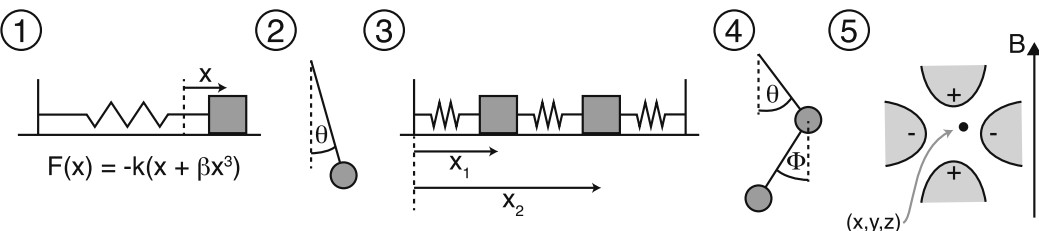

**Figure 1** Sketches of the five test systems that we consider.

is found that fits or a maximum bound on model complexity is reached. This heuristic algorithm only crudely captures the notion of mathematical complexity of the model, but it seems to work adequately well.

Note that, for a given polynomial model, it is possible to partially pre-calculate the score function $EL_D$ for a given dataset, yielding a function quadratic in the coefficients $c_i$. This is possible because the form of the model is fixed and it is possible to calculate its derivatives in advance. As a result, after the initial simplification of the score function, optimisation iterations are fast, and have a run-time independent of the number of data points.

Code for the score function, search algorithms and the datasets we use below can be downloaded from *Hudson, Hills & Grütter (a)*.

## RESULTS

We will consider five test systems, illustrated in Fig. 1. The first is the unforced Duffing oscillator, a textbook non-linear system. The second, a simple pendulum, is interesting because its Lagrangian cannot be represented exactly in the restricted polynomial representation. The third system, two masses on a frictionless surface joined by three springs to each other and two immovable walls, has two coupled degrees of freedom. The fourth system is the double pendulum, a coupled, two degree-of-freedom non-linear system capable of chaotic motion. As with the simple pendulum, the double pendulum cannot have its Lagrangian represented exactly by a finite degree polynomial. The fifth and final system is the Penning-type ion-trap, a three degree-of-freedom system with magnetic and electrostatic forces, that is of considerable experimental relevance.

Figure 2 shows the result of applying the algorithm to simulated data sets for these systems. It can be seen that the algorithm is able to successfully predict the future dynamics of all of the test systems. Let us look in detail at the progress of the algorithm, and the resulting learned models, for two of the example systems.

In the case of the Duffing oscillator the algorithm tried seven, increasingly complex, polynomial models to arrive at the prediction shown, which was generated by the model with $m = 4$ and $p = 4$. The final model has 10 free parameters, and required 2,160 Nelder–Mead iterations to optimise. The complete search working through all seven models, with a single-threaded implementation, executes in under five seconds on a 2012 2.0GHz Intel Core i7-3667U powered MacBook Air. The optimised Lagrangian, where we have removed terms with coefficients less than $10^{-5}$ and displayed the remaining

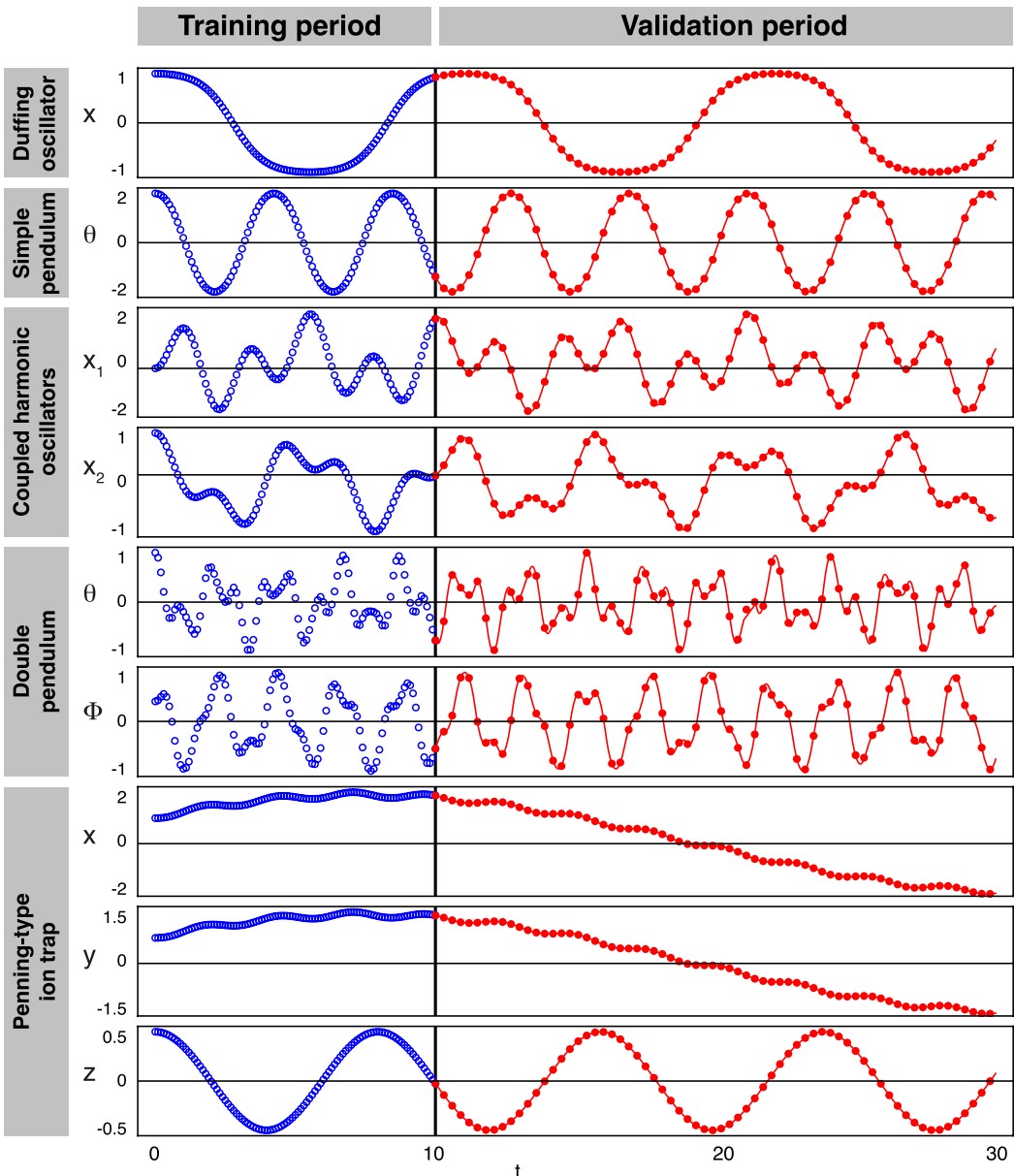

**Figure 2 Result of running the algorithm on simulated data from the five test systems.** In each graph panel, the (blue) open circles to the left of the vertical bar are the training data. The solid (red) line, to the right of the vertical bar is the algorithm's prediction. The (red) filled circles to the right of the bar show the actual behaviour of the system. For clarity only every third validation point is shown. The algorithm does not have access to these validation points when it is making its prediction. It can be seen that the algorithm has accurately learned the dynamics of the system in each case.

coefficients to two decimal places for clarity, was

$$\mathcal{L} = -0.30x^2 + 0.14x^4 + 0.20\dot{x}^2.$$

This is exactly the expression, apart perhaps from overall scaling, that would be written by a human physicist. The coefficients yielded by the search are found to match the correct

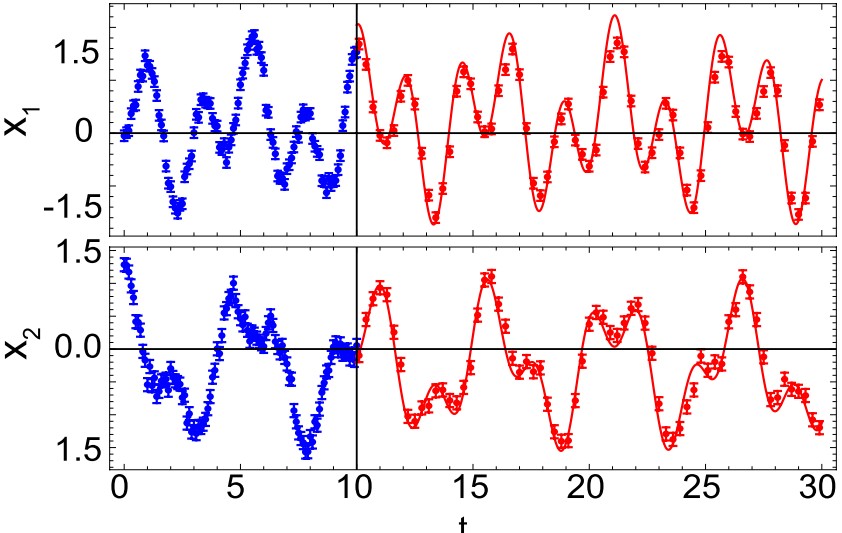

**Figure 3 Application of the algorithm to data with simulated noise added.** The graphs are in the same format as Fig. 2. We see that the algorithm is robust to noise, finding a model that accurately predicts the future evolution.

coefficients to the 6th decimal place, limited by the convergence tolerance that we set. By generating a model in this form the algorithm gives insight into the system directly from the data.

The case of the simple pendulum is also interesting to consider. Here the search algorithm tried three models, where the third, with $m = 2$ and $p = 4$, converged in 480 Nelder–Mead iterations. The search in this case took around 0.6 s. The generated model, multiplied by 100 to make it more readable, was

$$\mathcal{L} = 0.049x + 8.6x^2 - 3.0\dot{x}^2 - 0.0030x\dot{x}^2 - 0.41x^2\dot{x}^2. \tag{4}$$

It can be noted that this is not a straightforward Taylor expansion of the simple pendulum's Lagrangian, and it is not obvious how to relate it to the standard form. Experimenting with removing terms and solving the resultant equations of motion indicates that the terms proportional to $x$ and $x\dot{x}^2$ are unimportant, but the relatively small term in $x^2\dot{x}^2$ is essential. Despite being in an unexpected form, this Lagrangian does make successful predictions. We shall see in 'Generalisation' that it is in fact a local approximation of a true Lagrangian around the region of configuration space that the training trajectory explored.

Real world measurements are inevitably noisy and so to be practically useful it is important that the algorithm is able to converge even in the presence of imperfections in the data. We took the data for our third test system—the coupled harmonic oscillators—and added normally distributed noise, with standard deviation 0.1 (about 5% of the oscillation amplitude) to the position, velocity, and acceleration. Figure 3 shows the result of running the algorithm on this noisy data set. We see that the algorithm is robust to this noise, finding a model that describes the future evolution well.

**Table 1 Summary of models found for each of the test systems and the computational effort required to find them.** "Total iterations" gives the number of Nelder–Mead iterations used searching through all forms of the model, including the final form. "Final iterations" gives the number of Nelder–Mead iterations used refining the final form of the model. The "Generalised simple pendulum" will be introduced in 'Generalisation'.

| System | (m,p) | Parameters | Total iterations | Final iterations | Time to converge (s) |
|---|---|---|---|---|---|
| Duffing oscillator | (4,4) | 10 | 5,340 | 2,160 | 4.5 |
| Simple pendulum | (2,4) | 5 | 940 | 580 | 0.5 |
| Coupled harmonic oscillator | (2,2) | 10 | 1,740 | 1,740 | 1.2 |
| Double pendulum | (2,4) | 43 | 400,970 | 269,470 | 460 |
| Penning trap | (2,2) | 21 | 5,650 | 5,650 | 6.2 |
| Oscillator with noise | (2,2) | 10 | 1,240 | 1,240 | 0.7 |
| Generalised simple pendulum | (4,4) | 10 | 3,270 | 8,450 | 7.1 |

Table 1 summarises the size of the models found and the computational effort required to find them. We have not conducted a detailed study of how this proof-of-principle algorithm scales as the problem size increases, but note that our preliminary investigations show that it scales quite poorly. We find that the primary determinant of convergence rate seems to be the number of free parameters in the model. We therefore speculate that a more sophisticated technique for varying the structure of the model might be of use in improving the performance. A particularly promising approach might be to follow (*Clegg et al., 2005*) and use a genetic algorithm to evolve the form of the model in combination with a continuous algorithm like Nelder–Mead to optimise the parameter values for any given model form.

## GENERALISATION

We have shown that the algorithm can find models which successfully predict the future evolution of the system's behaviour. However, a good physical model does not just capture the behaviour of a particular time-series, corresponding to a particular set of initial conditions. Rather, it should be able to predict the behaviour of the system over a range of initial conditions. It is perhaps this ability to generalise that sets a true physical model apart from a mere fit or interpolation of the system's behaviour. It is interesting, therefore, to study whether the models found by our algorithm have this property.

We have seen in the case of the Duffing oscillator that the discovered model is indeed the correct model, and we would expect that this model will correctly predict the dynamics of the system for any initial conditions. We test this by simulating the behaviour of the system for a wide range of initial conditions, and comparing the results to the predictions of the model. We find, as expected, that the learned model for the Duffing oscillator does make correct predictions for all initial conditions.

Applying this procedure to the other test systems we find that the coupled harmonic oscillators and the Penning-type ion trap models also generalise well, making successful predictions for all initial conditions. This indicates that our algorithm is not merely a

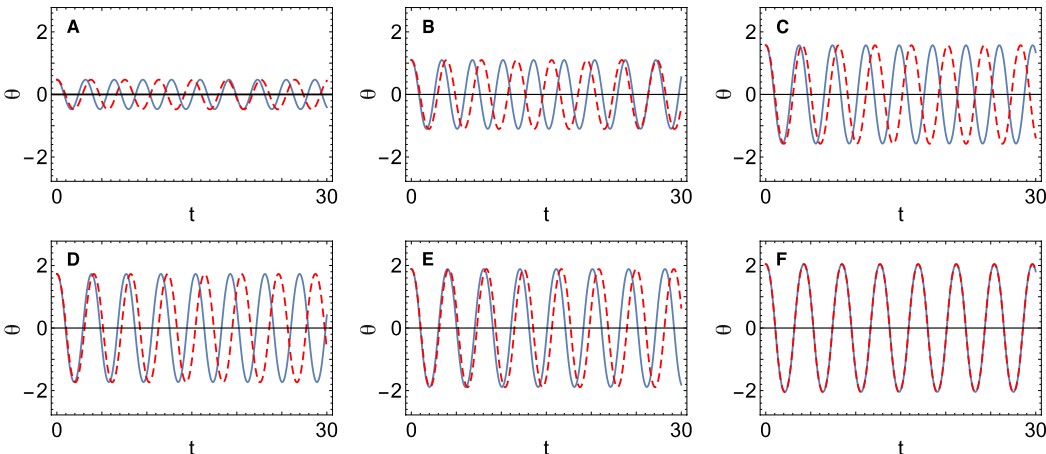

**Figure 4** **Predictions for different initial conditions of the learned simple pendulum model (red dashed line) compared to the true behaviour (blue solid line).** The amplitude of the pendulum swing varies between panels. The model was trained at the amplitude shown in (F). We see that the model makes good predictions for the initial conditions it was trained on, but breaks down for other initial conditions.

sophisticated curve fitting routine, but rather is finding the underlying physical truth behind the system dynamics to make its predictions.

The pendulum and double pendulum models do not generalise well, as we might have anticipated from the form of the Lagrangian in Eq. (4). Figure 4 compares the prediction of the learned simple pendulum model against the true behaviour, for a variety of swing amplitudes. We see that while the prediction is accurate for the amplitude at which the model was trained, it deviates at other amplitudes. These results are perhaps to be expected, and could well be the same as generated by a human physicist given the same data. The algorithm has found a mathematically simple approximation that works well for the data it has available to it, but does not have enough to go on to determine the true underlying model.

We consider two approaches to generating models that generalise better for these systems, inspired by the approaches a human physicist might take. The first method is simply to train the models with more data, corresponding to a wider range of initial conditions. The second is to introduce new mathematical constructs which allow a simpler model to be found, reasoning that this model is more likely to generalise well.

For the first approach we follow exactly the same procedure as before except we generate a number of trajectories, corresponding to a range of initial conditions, and use a score that is the sum of the scores for the individual trajectories. We applied this procedure to the simple pendulum system. The resulting search takes approximately 15 times longer to converge than the single-trajectory search. We find that the algorithm is unable to converge on an $m = 2$, $p = 4$ model, as it did before, and has to continue its search until it finds an $m = 4$, $p = 4$ model whose predictions fit all of the trajectories adequately. Figure 5 compares the predictive ability of this model with the 'single-trajectory' model of the previous section. We see that, as shown in Fig. 4, the single-trajectory model makes

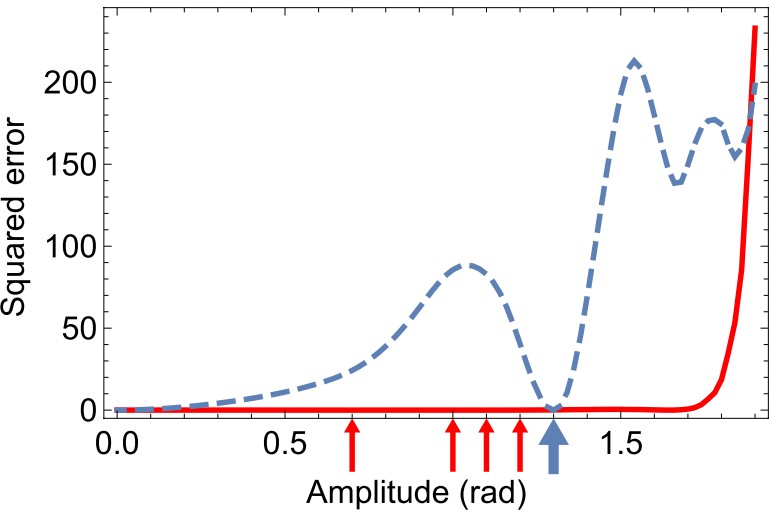

**Figure 5 Comparing a model for the simple pendulum trained on multiple trajectories with one trained on a single trajectory.** The curves show the squared error between the model's prediction and the true behaviour, as a function of the pendulum's swing amplitude. The dashed (blue) curve shows the result for a model trained at a single swing amplitude, indicated by the heavy (blue) arrow. This model performs well at the amplitude it was trained at, but poorly at other amplitudes. The model corresponding to the solid (red) curve was trained with multiple trajectories, indicated by the other (red) arrows. The original trajectory was also included in the training set for this model. We see that the 'multi-trajectory' model makes better predictions across a wide range of initial conditions, including conditions that it was not trained on. The multi-trajectory model is able to make successful predictions up to surprisingly large amplitudes, well beyond those it has seen in training.

good predictions for the initial condition it was trained at, but makes poor predictions for other initial conditions. The 'multi-trajectory' model, though, is much improved. It makes good predictions at all of the initial conditions it was trained at, and further makes good predictions at other, unseen initial conditions as well. We have found similar results for the double-pendulum system, although the computational expense of the problem constrained the experiment to a limited region of initial-condition-space.

Our second approach to generalisation is to expand the representation of the Lagrangians to encompass a wider range of mathematical expressions. We reason that, with a wider palette of mathematics at its disposal, the algorithm may be able to find a model of simpler form that works well. History has shown, although this may be tautological, that often systems of interest to physicists can be described by remarkably simple mathematical models. We hope that by allowing the algorithm to generate structurally simpler models, it may be more likely to discover the underlying physical truth.

We have developed a proof-of-principle implementation of a richer representation, and a corresponding search algorithm, detailed in the Appendix. Briefly, we take a genetic programming approach (*Koza, 1992*) and compose mathematical expressions as trees with leaf-nodes corresponding to the system variables, simple functions (sine, cosine, square) of these variables, and numerical constants. Branch-nodes of the tree are arithmetic operators $+, -, \times$. This structure can represent a much wider range of mathematical forms than our polynomial representation. We search over this tree-structured representation using

an algorithm (*Zitzler, Laumanns & Thiele, 2001*) that simultaneously tries to optimise the score and minimise the size of the trees. Thus, this search algorithm tries explicitly to find simple expressions that score well on the data.

Repeating the search on large-amplitude ($\pm0.95\pi$) simple pendulum data using the tree-based representation highlights the relative strength of this approach. The generated model, which makes a successful prediction, is

$$\mathcal{L} = 0.25\dot{\theta}^2 + 2.0\cos(\theta),$$

the same as would be written by a human physicist. Naturally, this model makes correct predictions over the full range of initial conditions. There are two reasons that the tree-based expression search is able to converge on this model. First it is only because the representation of possible models is richer that this model can be directly represented at all. Second, the notion of mathematical complexity in this representation more closely models that of a human physicist. This allows the search algorithm to do more work driving the result towards an expression that we recognize as canonical. It must be cautioned, though, that this is only a proof-of-principle demonstration. To reach this result we had to bias the search algorithm, as described in the Appendix, and even then the run time is significantly longer, often taking many hours with a multi-threaded implementation on the hardware described above. We were not able to get results for the double pendulum system at all with the computing resources at our disposal. Nonetheless, we present this result as the technique shows potential for learning models that are both better able to generalise, and in a format more suitable for communication to human physicists.

## CONCLUSION

We have demonstrated an algorithm that can predict the future dynamics of a physical system directly from observed data. We have shown that the algorithm generates models that can be communicated to a human physicist, sometimes even finding models in textbook form. We have further shown that the models generated generalise well to unseen data, and are not merely fits or interpolations, but are truly capturing the physical essence of the system under study.

One might ask what the use of such an algorithm is. As a first point, we find the question of whether a computer can do science to be fascinating in itself. Investigating the limits of a computer's ability in this regard educates us as to the strengths and weaknesses of our current scientific processes, and invites us to consider a different perspective on our scientific work.

But perhaps a more practical answer is that tools such as this could assist humans in their work. We see this assistance as coming in two forms. The first is simply automating the actions of a scientist so they can be applied to more data. Techniques that can be automatically applied to datasets, scanning for scientifically interesting features—in the case of the algorithm in this paper, for example, finding that there is a least action principle at work—may come to be a fruitful approach to generating unexpected scientific leads as we head into a data-dominated era. The second is opening up the techniques of science to

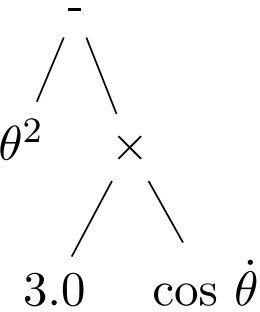

**Figure 6** An expression tree representing $\theta^2 - 3\cos(\dot{\theta})$.

non-specialists. By capturing the idea of searching for least action models in an algorithm we make it available to anyone, including those without the necessary skills to do it by hand. By way of analogy, it is interesting to consider popular online natural language translation software. While no-one would consider these tools suitable for translating poetry, they nonetheless are exceedingly useful to many people in the common case where a 'good-enough' translation will do. While we do not imagine computers will replace expert human physicists in the near term, we envisage the availability of tools to automate scientific reasoning will empower non-specialists to take better advantage of the discoveries and insights of physics.

## ACKNOWLEDGEMENTS

We acknowledge many useful discussions with Jeremy Mabbitt, Mike Tarbutt, Jack Devlin, Iain Barr, Ben Sauer, and Ed Hinds. DJAH and AMG contributed equally to this work.

## APPENDIX: TREE-BASED REPRESENTATION AND SEARCH

Here we describe in detail the tree-structured symbolic representation of mathematical expressions and the corresponding search routine. The expressions are built following a grammar designed to bias the search towards expressions that might be Lagrangians for simple physical systems. Each terminal of these expression trees is one of: numerical constants, randomly generated; the coordinate variables and velocities; the squares of the coordinates and velocities; and for coordinates which represent angles, the sine and cosine of the coordinates and velocities. The non-terminal nodes of the trees are the operators $+, -, \times$. An example expression tree is shown in Fig. 6, representing the expression $\theta^2 - 3\cos(\dot{\theta})$.

To search through these tree-structured expressions we take a genetic programming approach (*Koza, 1992*), explicitly optimising both the least-action score Eq. (3) and also a complexity score, using the SPEA2 multi-objective optimisation algorithm (*Zitzler, Laumanns & Thiele, 2001*). This biologically-inspired evolutionary algorithm maintains a population of candidate expressions and breeds, reproduces, and mutates them to try to simultaneously optimise the least-action and complexity scores.

In detail, we first construct a population of randomly generated expressions, usually numbering 100. We score these expressions using the least-action score and also assign a complexity score which is simply the number of nodes in the expression tree. The SPEA2 algorithm takes the current population, and an initially empty set of elite expressions, representing the best that have been seen so far. It has a rather complex selection mechanism (*Zitzler, Laumanns & Thiele, 2001*) that produces a new set of elite expressions, plus a set of expressions, the breeding pool, which are candidates for reproduction. A new generation is created from the breeding pool by mutation (10%) and pair-wise crossover (90%). Mutation is effected by replacing a randomly chosen subtree of the given expression with a randomly generated subtree. The crossover operation takes two expressions, selects a random point in each of the two trees, and swaps the sub-trees at these points to generate two new expressions. The evolutionary process is repeated starting from this new generation, and we iterate for a large number of generations, typically many thousand. To improve the convergence speed of the numeric constants in the expressions we also incorporate a small amount of hill-descent into each evolutionary iteration: a subset (20%) of the expressions have their numeric constants randomly adjusted by a small amount, and if this improves their least-action score, the modification is kept. We also impose a maximum size of expression (50 nodes) and trim expressions that exceed it each generation to ensure that the run-time is bounded. The final elite set is a set of expressions that represent the trade-off between least-action score and complexity. We select from this set the simplest expression that has a least-action error below a specified threshold.

In this tree-based method, the structure of the candidate Lagrangians varies during the search, so it is not possible to partially pre-calculate the score function, as it was in the restricted polynomial technique. Rather it must be calculated in full for each expression in the population. Further, the search space of possible expressions is exceedingly large, and the score is not very smooth with respect to the genetic operations. As a result, the search must run for many generations and is extremely computationally expensive. A naïve implementation might calculate the partial derivative time-series in (Eq. (3)) by symbolically differentiating the candidate expression and then calculating the value of the derivative. This, however, can be exponentially expensive in the depth of the expression, in terms of both memory and runtime. A better approach, that we adopt in this work, is to simultaneously evaluate the expression's value and its derivatives using automatic differentiation (*Kalman, 2002*). This method avoids calculating an expression for the derivative, and has run-time proportional to the size of the expression.

### Funding
The authors received no funding for this work.

### Competing Interests
The authors declare there are no competing interests.

## Author Contributions

- Daniel J.A. Hills and Adrian M. Grütter performed the experiments, analyzed the data, performed the computation work, reviewed drafts of the paper.
- Jonathan J. Hudson conceived and designed the experiments, performed the experiments, analyzed the data, contributed reagents/materials/analysis tools, wrote the paper, prepared figures and/or tables, performed the computation work, reviewed drafts of the paper.

## Data Availability

https://github.com/JonyEpsilon/flow.

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
