# Peer review of "An algorithm for discovering Lagrangians automatically from data"

_PeerJ Computer Science, doi:10.7717/peerj-cs.31_

## Round 0.1 · original submission · Minor Revisions

The above manuscript has been reviewed by two of our referees.
Comments from the reports appear below for your consideration.

·

Basic reporting

No comments

Experimental design

No comments

Validity of the findings

A formatting suggestion: Summarize results for test systems pertaining to final model (the (m,p) parameters, number of free parameters, computational time & number of iterations required) in a table.
On that note, there doesn't seem to be much analysis on the differing convergence rates for different systems - what conditions cause this-or-that test system to converge quicker?
Would also like to see some comments on how the computation time scales for 2-dimensional or higher dimensional systems (since every test system presented has been 1-dimensional) - is this approach for finding the Lagrangian computationally feasible for larger systems?

·

Basic reporting

No Comments

Experimental design

No Comments

Validity of the findings

No Comments

Additional comments

The idea of using the principle of least action seems novel and is interesting. The paper is well written, structured and complicated physics and algorithms are explained well albeit at a high level of abstraction.

I was left wondering a little about the scalability of the first approach using the Nelder-Mead optimisation. The examples, and the excellent results even with noise, seem to be dealing with situations with very small values of m and p. Could you say something about how this method would scale to more complicated examples? Also, what sorts of values of m and p might be found in typical real-world examples?

In the second, GA, approach the use of syntactic expressions as the genetic material is appropriate but I do not think that that is novel, as such. Perhaps some more references to related previous work is needed here.

Minor points... Some of the specifics of the modifications to EL_D to construct the score function seem a little arbitrary. For the non-physicist reader the discounting of certain terms from the Lagrangian because they can have "no physical significance" could be explained.

---

## Round 0.2 · accepted · Accept

Your submission with revision is now accepted for publication.